# Pain Catastrophizing in Cancer Patients

**DOI:** 10.3390/cancers16030568

**Published:** 2024-01-29

**Authors:** Sebastiano Mercadante, Patrizia Ferrera, Alessio Lo Cascio, Alessandra Casuccio

**Affiliations:** 1Main Regional Center of Pain Relief and Supportive/Palliative Care, La Maddalena Cancer Center, Via San Lorenzo 312, 90146 Palermo, Italylocascio.alessio@lamaddalenanet.it (A.L.C.); 2Department of Health Promotion, Maternal and Infant Care, Internal Medicine and Medical Specialties, University of Palermo, 90127 Palermo, Italy; alessandra.casuccio@unipa.it

**Keywords:** cancer pain, catastrophism, palliative care

## Abstract

**Simple Summary:**

Catastrophism was not associated with the levels of pain intensity, PSG, PSGR, and PGI for pain, except the rumination subscale that was independently associated with pain intensity at T0. A comprehensive palliative care management provided the relevant changes in symptom burden and annulled the pain expression associated with rumination.

**Abstract:**

Background: Pain catastrophizing is a group of negative irrational cognitions in the context of anticipated or actual pain. The aim of this study was to decipher the possible role of catastrophism on pain expression and outcomes after a comprehensive palliative care treatment. Methods: A consecutive sample of patients with uncontrolled pain was assessed. Demographic characteristics, symptom intensity included in the Edmonton symptom assessment system (ESAS), and opioid drugs used were recorded at admission (T0). The Pain Catastrophizing Scale (PCS) was measured for patients. Patients were also asked about their personalized symptom goal (PSG) for each symptom of ESAS. One week after a comprehensive palliative care treatment (T7), ESAS and opioid doses used were recorded again, and the number of patients who achieved their PSG (PSGR) were calculated. At the same interval (T7), Minimal Clinically Important Difference (MCID) was calculated using patient global impression (PGI). Results: Ninety-five patients were eligible. A significant decrease in symptom intensity was reported for all ESAS items. PGI was positive for all symptoms, with higher values for pain, poor well-being, and poor sleep. Only the rumination subscale of catastrophism was significantly associated with pain at T0 (B = 0.540; *p* = 0.034). Conclusions: Catastrophism was not associated with the levels of pain intensity, PSG, PSGR, and PGI for pain, except the rumination subscale that was associated with pain intensity at T0. A comprehensive palliative care management provided the relevant changes in symptom burden, undoing the pain expression associated with rumination.

## 1. Introduction

Approximately 40% to 90% of cancer patients experience pain, with more than 30% reporting moderate–severe pain [1]. Pain affects many aspects of patient lives, including functional, psychological, and social well-being dimensions. The cancer population experiences psychosocial distress after their diagnosis and during cancer treatment, possibly exacerbating their pain [2,3,4]. Psychosocial variables may elucidate factors influencing the expression of pain. Identifying the relative contributions of modifiable psychosocial factors to cancer pain symptoms and opioid use could be helpful to suggest patient-centered cancer pain interventions [4].

Catastrophizing has been conceived as an exaggerated negative “mental set” brought about while bearing painful conditions. Pain catastrophizing is a group of negative irrational cognitions in the context of anticipated or actual pain [5,6]. It is characterized by inadaptable dimensions including rumination (the continuous negative thinking of pain), magnification (exaggerating the potential destructive power of pain), and helplessness (the perception of inability to cope with pain). It constitutes a relevant psychosocial factor in predicting the adjustment to chronic pain. The need to assess patients’ sociodemographic and psychosocial profiles when treating cancer pain has been emphasized, as they could be related to clinically meaningful differences in pain and the use of analgesics [5]. It has been recently reported in cancer and non-cancer patients that increasing pain catastrophism may predict the worsened pain severity [7]. Moreover, catastrophizing may impact the prescription and high-dose opioid use, likely due to beliefs about the appropriateness of pain medication [8]. However, in these studies, different populations have been examined, and the assessment was not performed in a specialistic palliative care setting, allowing a comprehensive palliative care treatment.

The Edmonton symptom assessment system (ESAS) is a unidimensional numeric rating scale used ubiquitously to evaluate the intensity of physical and psychological symptoms [9]. This tool may have some limitations because of its subjectivity. Personalized symptom goal (PSG) typically reflects an individual measure of outcome and provides a simple and individualized therapeutic “target” for each symptom [10,11,12,13,14,15]. Patient’s global impression (PGI) is a validated global rating-of-change scale used to assess the subjective patients’ response based on the individual feeling of improvement or deterioration after receiving a treatment [16,17]. These tools are considered useful for assessing patient-related outcome measures.

Information about the role of catastrophism in the pain and symptom management of patients with advanced cancer is lacking. The hypothesis of this study was that catastrophism may influence pain expression. The primary outcome was the assessment of the catastrophizing and pain expression of pain in cancer patients admitted to an acute supportive/palliative care unit (ASPCU). The secondary outcome was to examine the eventual influences on clinical outcomes, in terms of changes in ESAS items, PGI, and the achievement of PSG response (PSGR) after one week of comprehensive palliative care treatment.

## 2. Materials and Methods

A consecutive sample of patients admitted to the ASPCU from September 2022 to June 2023 was enrolled. The protocol was approved by the ethical committee of the University of Palermo. All patients provided written informed consent. Inclusion criteria were age ≥18 years, a diagnosis of advanced cancer, defined as a relapse, a metastasis, or a local advanced disease, and pain intensity of ≥4/10. Patients with a lower level of pain intensity were likely to not need changes in their analgesic therapy. Exclusion criteria were incapacity to complete the questions due to cognitive or linguistic problems, an expected survival of ≤2 weeks, or a Karnofsly level of ≤30.

### 2.1. Measurements

Demographic characteristics, including age, gender, primary cancer diagnosis, and the Karnosky level were recorded. Karnofsky is a well-known parameter for measuring the level of physical activity in cancer patients. ESAS (Edmonton symptom assessment system) items and opioid drugs used were recorded at admission (T0) and after one week of comprehensive palliative care treatment (T7). Opioid doses were expressed as oral morphine equivalents (OME) [18]. ESAS is a tool assessing the intensity of the most common psychological and physical symptoms, which is valid and reliable for assessing the global symptom burden [19]. The Pain Catastrophizing Scale (PCS) was measured at T0 and T7. PCS consists of 13 items [20], rated on a 5-point Likert scale, from 0 (not at all) to 4 (all the time). A total score is computed by summing the 13 items, with higher scores indicating a greater pain catastrophizing. The PCS has demonstrated reliability and validity and has been widely used in patients with chronic pain globally [21,22,23]. Patients were dichotomized according to their PCS (<30 and ≥30) [20,24].

Patients were also asked about their personalized symptom goal (PSG) for each symptom of ESAS. The question was “At what level would you feel comfortable using the 0–10 numeric rating scale used for ESAS?” [12,13]. All patients underwent comprehensive and continuous symptom assessment and management targeted to individual characteristics during their hospital stay at the ASPCU. One week after a comprehensive palliative care treatment (T7), ESAS and opioid doses used were recorded again, and the number of patients who achieved their PSG (PSGR) was calculated. At the same interval (T7), Minimal Clinically Important Difference (MCID) was calculated using the patient global impression (PGI) of improvement according to the following scale: 3 = much better; 2 = better; 1 = a bit better; 0 = the same; −1 = a little worse; −2 = worse; and −3 = much worse [12,13]. PGI has been used as an anchor for evaluating a clinically significant change in symptom intensity [11]. Data reporting followed the STROBE checklist [25].

### 2.2. Statistics

The statistical analysis of quantitative and qualitative variables, including descriptive statistics, was performed for all items. Discrete and continuous variables were reported in terms of frequency (%) means (± SD) and medians (interquartile range, IQR). The normality of the distribution of continuous variables was assessed using the Shapiro–Wilk test.

The paired *t*-test was used to evaluate the intragroup symptom changes and opioid doses at the two time intervals. The independent Student *t*-test was used to evaluate the clinical variable differences between two pain catastrophizing patient groups. The correlation between predictor variables as pain catastrophism subscales and outcome variables (the changes in symptom severity, PGS, and PGI) were evaluated using univariable and multivariable linear regression models. B values and related 95% confidence intervals (95% CI) were reported as well as the *p*-value. Data were analyzed using the IBM SPSS Software 24 version (IBM Corp., Armonk, NY, USA). All *p*-values were two-sided, and *p* ≤ 0.05 was considered statistically significant.

## 3. Results

Three-hundred-eighty-two patients were screened in the period taken into consideration. Of them, 287 patients were deemed ineligible: 37 did not provide informed consent, 218 reported a pain intensity of less than four, 55 presented a neurological impairment, 39 were either illiterate or faced a language barrier, and 28 had a life expectancy of less than two weeks (Figure 1).

Ninety-five were found to be eligible for this study. The mean age was 61.9 years (SD = 11.5), and fifty-three were females. The mean Karnofsky status was 53.9 (SD = 9). The most frequent primary diagnosis was gastrointestinal. Details and other characteristics are reported in Table 1. No differences between patients with low and high PCS were found.

Data regarding the changes in ESAS items and OME in patients with low and high PCS are reported in Table 2. Significant decreases in symptom intensity were found after 1 week of comprehensive palliative care in both groups, particularly in total ESAS, that represents the global psychological and physical distress score. There were intergroup differences for anxiety, depression, weakness, poor well-being, and total ESAS at both T0 (admission) and T7 (one week after). Values of OME at T0 were significantly higher in patients with high PCS (≥30) at T0, but these differences disappeared after 1 week of comprehensive palliative care (T7).

Changes in PSG, PSGR, and PGI after one week of comprehensive palliative care are reported in Table 3. There were significant differences in PSG for depression and poor appetite between patients with low PCR (<30) and high PCR (≥30). There were also significant differences in PGI for poor sleep between the two groups. Significant differences were also found in PSGR for some items, including dyspnea, depression, poor well-being, and particularly anxiety.

PCS and details on its subscales of magnification, rumination, and helplessness are shown in Table 4. In the univariable linear regression analysis, only pain intensity at T0 was correlated with the rumination subscale (B = 0.497 (95% CI = 0.014–0.980); *p* = 0.044), while no association between PGI, PSG, and PSGR for pain, as well as other demographic variables, and PCS was found. Rumination was found to be independently associated with pain intensity at T0 (B = 0.540 (95% CI = 0.040–0.098); *p* = 0.034) in the multivariable linear regression model analysis.

## 4. Discussion

This study reported interesting findings regarding the influence of a negative mental set like catastrophism in patients with uncontrolled cancer pain admitted to the ASPCU. The principal finding was that catastrophism was not associated with the levels of pain intensity, expectations (PSG), the achievement of expectations (PSGR), and the individual impression of clinical improvement (PGI) for pain. Indeed, the rumination subscale was independently associated with pain intensity at T0. Interestingly, such an association was not found at T7, one week after starting comprehensive palliative care. Pain and other symptoms included in ESAS significantly changed after a week of comprehensive palliative care treatment. The improvement in analgesia was achieved without an increase in opioid dosage, as OME did not change. These data are consistent with data reported in a similar ASPCU, where half of the patients achieved clinically improved pain without an OME increase, suggesting that a multidimensional palliative care intervention is effective in improving pain control in many patients without the need to increase the opioid dose, possibly as a consequence of careful opioid switching and individual dosing [26]. These data are confirmed by the achievement of PSGR for pain in a large number of patients (58.9%). Thus, specialized palliative care provided in the ASPCU plays a fundamental role in the effective short-term pain control, as it provides not only the achievement of effective analgesia with an adequate optimization of opioid use but also psychological support and intensive management of other symptoms or opioid-related adverse effects.

Pain catastrophism is conceptualized as a negative cognitive–affective response to the anticipated or actual pain and has been associated with several important pain-related outcomes [27]. Despite the very low quality of the available evidence, the general consistency of the findings highlights the potential role that catastrophism may play in delaying recovery from chronic non-malignant pain [8,28].

There is a paucity of data assessing catastrophism in patients with cancer pain. In a longitudinal retrospective study of cancer patients with chronic pain who completed the follow-up performed at about 5 months, a significant decrease in pain severity, but not interference, was reported. Increased pain catastrophizing was significantly associated with increased pain severity over time. Pain catastrophizing and increased depression were significantly associated with increased pain interference over time. These observations indicate that cancer patients with chronic pain would likely benefit from the incorporation of nonpharmacological interventions, simultaneously addressing pain and psychological symptoms [7]. A retrospective cross-sectional study of chronic and cancer pain assessed the associations between biopsychosocial factors and pain and opioid use among individuals with chronic pain cancer. The modifiable psychological factors, including sleep disturbances, depression, and pain catastrophizing, were associated with pain and opioid use in patients with chronic pain [5]. However, treatment was not performed in a specialistic palliative care setting, where a comprehensive care treatment may mitigate the possible psychological influence. The study was completed in half of the patients, and the study data were entered into a computerized platform. This is also confirmed by the limited changes in pain intensity at follow-up, the timing of which was unclear.

Some more information is available on the influence of psychological symptoms on pain expression in patients with advanced cancer, although data are controversial because of the different methodologies and tools used [29,30,31,32,33,34]. In 397 cancer patients, anxiety and depression had strong and independent associations with mental health domains and somatic symptom burden. Depression had a more pervasive association with the multiple other domains of health-related quality of life [29]. In a retrospective study of 216 patients, those with depressive mood expressed a higher frequency of drowsiness nausea, pain, dyspnea, poor appetite, and poor well-being and expressed a higher intensity of symptoms. Patients with anxiety expressed a higher frequency of nausea, pain, and dyspnea, and expressed a higher intensity of pain, fatigue, poor appetite, and poor well-being [30]. In a secondary analysis of cross-sectional data of 487 patients with advanced cancer, depression severity significantly correlated with the number of physical symptoms, symptom distress, and symptom severity [31] In a multicenter observational study, depression was associated with significantly higher scores of symptoms [32]. In a study of eighty-seven patients, depression was found to be an independent predictor of poor survival in patients with advanced cancer [33]. Finally, in a secondary analysis of a cross-sectional study of 2768 outpatients, significant pain was found to be independently associated with emotional distress [34]. Most studies lack longitudinal observation, without a second point assessment. In a longitudinal study, even after a palliative care intervention, able to produce a significant decrease in the intensity of ESAS symptoms, anxiety and depression were still associated with symptom hyper-expression. Thus, anxiety and depression may affect the expression of most symptoms included in the ESAS [35]. Indeed, psychological distress, including depression and anxiety, is not equivalent to the concept of catastrophism that has a different construct.

This study has few limitations. The reasons for a lack of significant associations between PCS and measured pain parameters could be due to the exclusion criteria, for example, short survival or low performance status or the level of pain intensity. This was a single center study performed in an ASPCU, with advanced knowledge and experience in providing expert comprehensive treatment. To optimize the clinical outcome, this approach requires to be highly individualized and cannot entail a strict protocol.

## 5. Conclusions

Catastrophism was not associated with the levels of pain intensity, PSG, PSGR, and PGI for pain, except the rumination subscale that was independently associated with pain intensity at T0. A comprehensive palliative care management provided relevant changes in symptom burden and annulled pain expression associated with rumination.

Further studies in different settings with a large number of patients could be useful to confirm the preliminary findings of this study and to clarify the relationship between catastrophism and pain in patients with advanced cancer.

## Figures and Tables

**Figure 1 cancers-16-00568-f001:**
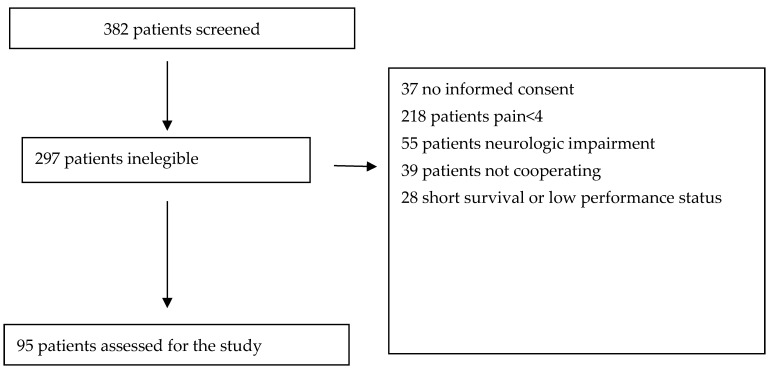
Patients’ flow chart.

**Table 1 cancers-16-00568-t001:** Epidemiological, clinical, and sociocultural characteristics of patients with low PCS (<30) and high PCS (≥30).

	Overall N° 95	PCS < 30; N° 57	PCS ≥ 30; N° 38	*p*
Age (years, mean, ±SD)	61.9 (11.5)	62.2(11.1)	61.4(12.1)	0.753
Gender (female/male)	53/42	32/25	21/17	0.933
Karnofsky (mean SD)	53.9 (9.0)	53.9(9.2)	53.9(8.8)	0.963
Primary tumor N°				
Gastrointestinal	29	21	8	
Breast	23	12	11	
Lung	23	15	8	0.338
Genitourinary	14	6	8	
Others	6	3	3	
Education N°				
No education	1	1	0	
Primary	22	14	8	0.964
Secondary school	39	22	17	
High school	24	15	9	
Degree	9	5	4	
Believer N°	51	33	18	
Practicing believer	41	23	18	0.510
Not believer	3	1	2	

**Table 2 cancers-16-00568-t002:** Changes (mean, ±SD) in the intensity of ESAS (Edmonton symptom assessment system) items at admission (T0) and after one week (T7) of comprehensive palliative care in patients with low and high PCS. OME = oral morphine equivalents.

	PCS < 30		PCS ≥ 30	
	T0	T7	T0	T7
Pain	6.1 (1.9)	2.0 (1.6) **	6.8 (1.9)	2.4 (1.7) **
Dyspnea	0.7 (2.3)	0.0 (0.0) *	0.7 (2.1)	0.4 (1.2) °
Anxiety	2.9 (3.1)	1.1 (1.9) **	4.6 (3.5) °	2.7 (2.6) **°
Depression	2.5 (3.3)	0.6 (1.5) **	4.4 (3.6) °	2.3 (2.7) **°
Poor sleep	3.3 (3.2)	1.6 (2.6) **	4.6 (3.7)	1.7 (2.0) **
Drowsiness	2.8 (2.6)	1.6 (2.0) **	4.3 (2.7) °	2.6 (2.7) **
Nausea	1.2 (2.7)	0.5 (1.5)	1.2 (2.5)	0.8 (2.2)
Poor appetite	2.3 (3.4)	1.3 (2.2)	4.5 (3.7) °	2.6 (2.9) **°
Weakness	4.8 (3.2)	2.3 (2.5) **	6.6 (2.3) °	3.5 (3.0) **°
Poor well-being	4.6 (3.2)	1.5 (2.0) **	6.6 (2.6) °	3.1 (2.9) **°
Total ESAS	31.9 (14.7)	13.1 (11.0) **	45.5 (16.9) °	23.61 (16.8) **°
OME	78.8 (82)	93.4 (88)	176.7 (209) °	122.1 (98)

Intragroup difference (paired *t*-test): * = *p* < 0.05 and ** = *p* < 0.001. Intergroup difference (independent Student *t*-test): ° = *p* < 0.01.

**Table 3 cancers-16-00568-t003:** Changes in personalized symptom goal (PSG) (mean, SD) and personalized symptom goal response (PSGR) (the number and percentage of patients). Patient global impression (PGI) (mean, SD) after one week of comprehensive palliative care. Intergroup difference: * = *p* < 0.05 and ** = *p* < 0.001.

	PCR < 30	PCR ≥ 30	PCR < 30	PCR ≥ 30	PC < 30	PC ≥ 30
	PSG	PSG	PGI	PGI	PSGR	PSGR
Pain	2.0 (2.0)	2.3 (2.0)	2.0 (1.1)	1.7 (1.1)	34 (59.6%)	22 (57.9%)
Dyspnea	0.04 (0.3)	0.2 (0.6)	0.4 (0.9)	0.4 (0.9)	57 (100%)	34 (89.5%) *
Anxiety	0.8 (1.5)	1.2 (1.7)	0.6 (0.9)	0.2 (1.1)	42 (73.7%)	17 (44.7%) **
Depression	0.5 (1.3)	1.2 (1.8) *	0.6 (0.9)	0.5 (1.2)	49 (86.0%)	25 (65.8%) *
Poor sleep	0.9 (1.7)	1.0 (1.9)	0.7 (1.2)	1.2 (1.3) *	40 (70.2%)	24 (63.2%)
Drowsiness	0.9 (1.6)	1.6 (1.9)	0.4 (1.0)	0.2 (1.2)	34 (59.6%)	19 (50.0%)
Nausea	0.0 (0.0)	0.1 (0.5)	0.4 (0.9)	0.6 (1.2)	50 (87.7%)	32 (84.2%)
Poor appetite	0.7 (1.4)	1.7 (2.0) **	0.7 (1.4)	0.2 (1.3)	40 (70.2%)	26 (68.4%)
Weakness	1.3 (1.7)	2.1 (2.0)	1.0 (1.2)	0.6 (1.5)	36 (63.2%)	18 (47.4%)
Poor well-being	1.2 (1.7)	1.5 (1.8)	1.2 (1.0)	1.1 (1.2)	39 (68.4%)	18 (47.4%) *

**Table 4 cancers-16-00568-t004:** Pain Catastrophizing Scale and its subscales. Values are expressed as means (SDs) and medians (interquartile ranges, IQRs).

	Overall	PC < 30; N° 57	PC ≥ 30; N° 38
Pain Catastrophizing Scale			
(mean SD)	26.9 (10.5)	20.1 (6.9)	37.2 (5.3)
Median (IQR)	27(19–34)	21(15–26)	36(33–41)
Magnification subscale			
(mean SD)	3.5 (2.3)	2.7 (2.2)	4.8 (2.0)
Median (IQR)	3(2–5)	2(1–4)	5(4–6)
Rumination subscale			
(mean SD)	12.7 (4.6)	10.5 (4.4)	16.1 (2.4)
Median (IQR)	13(10–16)	11(8–13)	16(14–18)
Helplessness subscale			
(mean SD)	10.7 (5.7)	7.4 (3.8)	15.8 (4.0)
Median (IQR)	11(7–14)	8(4–10)	15(13–19)

## Data Availability

Data are available on request.

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
