# Peer review of "Pain Catastrophizing in Cancer Patients"

_cancers, 2024, doi:10.3390/cancers16030568_

Round 1
Reviewer 1 Report
Comments and Suggestions for Authors
The topic of this study is attractive because it is obvious that the pain's 'catastrophizing nature' worsens all symptoms, especially in palliative care when patients are experiencing many anxiety factors.
Unfortunately, the study is subject to significant inaccuracies and limitations.
1 – Methodology
For Pain catastrophizing symptoms scale (PCS), there is no cut-off level in scale between patients who have “catastrophizing symptoms” and others. Thus, it’s challenging to really evaluate the incidence and importance of these symptoms in palliative population as “The aim of this study was to assess catastrophizing and the possible role in expression of pain in patients with cancer admitted to an acute supportive/palliative care unit (ASPCU)”. You would have to compare patients with a low level of catastrophizing symptoms and those with a high (significant) level. It’s not clear in this study if you proceeded to a PCS evaluation at T0 and T7. Without these evaluations any analyze is challenging.
Additionally, your study does not provide any information on comprehensive palliative care management, you need to specify a description.
The Institutional Review Board report must be provided in attached file for such a study.
2 Results
The description is quite short as the larger part is the flow chart and characteristics of patients. Moreover, outcomes are quite difficult to understand.
Table 2: the values at T0 and T7 are included in the first part, we lack understanding of what PSG, PSGR, and PGI mean in front of each symptom. Is there a distinction between patients with a high Pain Catastrophizing score and those with a low one?
Your results are not understandable and cannot be interpreted.
Table 3: PCS is your main goal and you only provide this table with no data at T0 and T7
3 Discussion
L141: The principal finding was that catastrophism was not associated with the levels of pain intensity, PSG, PSGR, and PGI for pain. It’s quite challenging to understand your statement in front of your outcomes.
Limitations
L201- 202 : It is a little bit exaggerated to claim that one limitation is the patients management in an expert ASPCU. It is humiliating for others. Moreover, what about the criteria to have “advance knowledge and experience in providing expert comprehensive treatments” in an ASPCU?
Author Response
The topic of this study is attractive because it is obvious that the pain's 'catastrophizing nature' worsens all symptoms, especially in palliative care when patients are experiencing many anxiety factors.
Unfortunately, the study is subject to significant inaccuracies and limitations.
1 – Methodology
For Pain catastrophizing symptoms scale (PCS), there is no cut-off level in scale between patients who have “catastrophizing symptoms” and others. Thus, it’s challenging to really evaluate the incidence and importance of these symptoms in palliative population as “The aim of this study was to assess catastrophizing and the possible role in expression of pain in patients with cancer admitted to an acute supportive/palliative care unit (ASPCU)”. You would have to compare patients with a low level of catastrophizing symptoms and those with a high (significant) level. It’s not clear in this study if you proceeded to a PCS evaluation at T0 and T7. Without these evaluations any analyze is challenging.
Additionally, your study does not provide any information on comprehensive palliative care management, you need to specify a description.
The Institutional Review Board report must be provided in attached file for such a study.
We added a cut-off suggested by literature. The paper was restructured according to this approach
PCS was measured at T0 and T7
Comprehensive treatment is quite individualized, according to individual needs, not based on specific protocol. Substantially is based on controlling associated symptoms or opioid-related problems, other than psychological support with a typical palliative care approach.
We added also the ethical committee approval
2 Results
The description is quite short as the larger part is the flow chart and characteristics of patients. Moreover, outcomes are quite difficult to understand.
Table 2: the values at T0 and T7 are included in the first part, we lack understanding of what PSG, PSGR, and PGI mean in front of each symptom. Is there a distinction between patients with a high Pain Catastrophizing score and those with a low one?
Your results are not understandable and cannot be interpreted.
Table 3: PCS is your main goal and you only provide this table with no data at T0 and T7
PSG, PSGR, and PGI are described in methods, and have been proposed as reliable PROMS in the large existing literature (see references). For each symptom these parameters were measure (not only for pain…). We also added an analysis based on cut-off
3 Discussion
L141: The principal finding was that catastrophism was not associated with the levels of pain intensity, PSG, PSGR, and PGI for pain. It’s quite challenging to understand your statement in front of your outcomes.
It is reported in discussion
Limitations
L201- 202 : It is a little bit exaggerated to claim that one limitation is the patients management in an expert ASPCU. It is humiliating for others. Moreover, what about the criteria to have “advance knowledge and experience in providing expert comprehensive treatments” in an ASPCU?
This statement is not humiliating, but simply reporting the expertise of this group, not repeatable in other settings (see references for APCU). This to me seems a limitation rather than a humiliation.
Reviewer 2 Report
Comments and Suggestions for Authors
I suggest the inclusion of other keywords such as: pain management
The authors of the STROBE checklist mentioned in “Materials and Methods” should be presented in the reference list.
Tables should be directly readable without the need for the manuscript. In table 1, are the values absolute or relative? I suggest to add N and (%). In the title I suggest adding at the end: (n=95).
In Table 2, the authors should insert a legend for the abbreviations T0, T7, ESAS, PSG, SD, PSGR, PGI, OME.
Ref 17- The publication year is not correct. It is 2006.
Almost 30% of the articles (n=11) were published more than 10 years ago, and only 2 from these were on validation scales.
Author Response
I suggest the inclusion of other keywords such as: pain management
The authors of the STROBE checklist mentioned in “Materials and Methods” should be presented in the reference list.
Added
Tables should be directly readable without the need for the manuscript. In table 1, are the values absolute or relative? I suggest to add N and (%). In the title I suggest adding at the end: (n=95).
According to suggestions of refree 1, patients were divided in two groups. Numbers are reported in table 1. In
Table 2, the authors should insert a legend for the abbreviations T0, T7, ESAS, PSG, SD, PSGR, PGI, OME.
Done
Ref 17- The publication year is not correct. It is 2006.
correct
Almost 30% of the articles (n=11) were published more than 10 years ago, and only 2 from these were on validation scales.
The meaning of this question is unclear
Reviewer 3 Report
Comments and Suggestions for Authors
Dear authors.
I have reviewed the manuscript "Pain Catastrophing in Patients With Cancer", ID cancers-2721095.
This is a well-designed observational study with a one-week longitudinal follow-up to determine a possible association between Pain Catastrophic Scale variables and baseline pain intensity and symptom control outcomes.
The abstract adequately summarises the aims, results and conclusions.
The background is well developed. The main instruments used in the study are adequately described in the introduction. The aims are summarised. However, the authors have not clearly stated the hypothesis of the study. I understand that the study aims to prove or reject the association between the Pain Catastrophic Scale and pain intensity and symptom control outcomes.
The methodology and statistical analysis are clear and well described.
The results are clearly presented. Tables adequately summarise the reported results. The lack of association between the Catastrophic Pain Scale and baseline pain intensity and symptom control outcomes is confirmed by univariable linear regression analysis for all subscales except rumination.
However, only the central measures (mean and standard deviation) of the grouped subscales of the Catastrophic Pain Scale (rumination, magnification and helplessness) were reported, which were sufficient to perform univariable linear regression. In my opinion, it would be interesting to add descriptive information on how many patients scored with at least moderate intensity on the 13 variables of the scale, in order to better understand the study population.
The morphine oral equivalent dose is defined as a variable for this study. In fact, the authors confirmed no changes in the morphine oral equivalent dose after 1 week of follow-up. However, the morphine oral equivalent dose is not reported in the results section.
Discussion arguments based on observed data and literature review are well developed. As I have already commented, the oral morphine equivalent dose is discussed in the discussion, but there is no information on this variable in the results section.
In my opinion, this manuscript should be accepted for publication with minor changes.
Author Response
I have reviewed the manuscript "Pain Catastrophing in Patients With Cancer", ID cancers-2721095.
This is a well-designed observational study with a one-week longitudinal follow-up to determine a possible association between Pain Catastrophic Scale variables and baseline pain intensity and symptom control outcomes.
The abstract adequately summarises the aims, results and conclusions.
The background is well developed. The main instruments used in the study are adequately described in the introduction. The aims are summarised. However, the authors have not clearly stated the hypothesis of the study. I understand that the study aims to prove or reject the association between the Pain Catastrophic Scale and pain intensity and symptom control outcomes.
The methodology and statistical analysis are clear and well described.
The results are clearly presented. Tables adequately summarise the reported results. The lack of association between the Catastrophic Pain Scale and baseline pain intensity and symptom control outcomes is confirmed by univariable linear regression analysis for all subscales except rumination.
However, only the central measures (mean and standard deviation) of the grouped subscales of the Catastrophic Pain Scale (rumination, magnification and helplessness) were reported, which were sufficient to perform univariable linear regression. In my opinion, it would be interesting to add descriptive information on how many patients scored with at least moderate intensity on the 13 variables of the scale, in order to better understand the study population.
I added this point
The morphine oral equivalent dose is defined as a variable for this study. In fact, the authors confirmed no changes in the morphine oral equivalent dose after 1 week of follow-up. However, the morphine oral equivalent dose is not reported in the results section.
It is in Table and now reported in the text
Discussion arguments based on observed data and literature review are well developed. As I have already commented, the oral morphine equivalent dose is discussed in the discussion, but there is no information on this variable in the results section.
In my opinion, this manuscript should be accepted for publication with minor changes
Reviewer 4 Report
Comments and Suggestions for Authors
The authors try to show the role of pain catastrophizing in cancer pain. The study hypothesis and the primary outcome are not clear. I do not understand what the authors tried to study.
Major comments
#1. The study hypothesis and the primary outcome are not clear (Page 2, Line 66). The authors need to describe the study hypothesis and the primary outcome more precisely.
#2. Ethical approval number is missing. The ethical issue should be first part of methods section.
#3. Why did the authors exclude pain intensity less than 3/10 (Page 2, Line 77)?
#4. I do not see data about univariate and multivariate analysis (Page 3, Line 110).
#5. There are quite some odd wordings and use of grammar. I do suggest having some external language editing done by a person familiar with the field. Also, the manuscript does not respect the elementary rules of a scientific writing.
Minor comments
#1. The authors show data as mean (SD). Were distributions of data all normal (Gaussian)?
#2. What is “Karnofsky” (Table 1)?
#3. There is no baseline characteristics like sex, height, weight, and so on (Table 1).
#4. I do not understand Table 1. What are Education, No education, Primary, Secondary school, Degree, Believer, Believer-practing, and Not believer?
#5. I do not understand from what tests these p values were calculated (Table 2).
Comments on the Quality of English Language
English very difficult to understand.
Author Response
The authors try to show the role of pain catastrophizing in cancer pain. The study hypothesis and the primary outcome are not clear. I do not understand what the authors tried to study.
Major comments
#1. The study hypothesis and the primary outcome are not clear (Page 2, Line 66). The authors need to describe the study hypothesis and the primary outcome more precisely.
The hypothesis is described with the aim at the end of introduction
#2. Ethical approval number is missing. The ethical issue should be first part of methods section.
added
#3. Why did the authors exclude pain intensity less than 3/10 (Page 2, Line 77)?
It is because generably this group of patients do not required changes in pain therapy
#4. I do not see data about univariate and multivariate analysis (Page 3, Line 110).
I added univariate and multivariate analysis
#5. There are quite some odd wordings and use of grammar. I do suggest having some external language editing done by a person familiar with the field. Also, the manuscript does not respect the elementary rules of a scientific writing.
Minor comments
#1. The authors show data as mean (SD). Were distributions of data all normal (Gaussian)?
Yes, The normality distribution of continuous variables was assessed by the Shapiro–Wilk test. However, for the pain catastrophing scale e subscale variables we have added in table 4 the medians and interquartile range values.
#2. What is “Karnofsky” (Table 1)?
Karnofsky is well know toll for measuring the level of physical activity in cancer patients
#3. There is no baseline characteristics like sex, height, weight, and so on (Table 1).
Height and weight are not usually reported in such kind of study. Gender was added
#4. I do not understand Table 1. What are Education, No education, Primary, Secondary school, Degree, Believer, Believer-practing, and Not believer?
I readapted the table for a better understanding. Believer and so on are referred to religiosity
#5. I do not understand from what tests these p values were calculated (Table 2).
I have added in the Table 2 the statistical test used
Round 2
Reviewer 3 Report
Comments and Suggestions for Authors
I have reviewed the changes to the manuscript.
I agree with the changes made and I think it can be accepted.
Author Response
Thanks so much for your help
Reviewer 4 Report
Comments and Suggestions for Authors
Comments and Suggestions for Authors
The authors do not respond my first comment. The study hypothesis and the primary outcome are not clear. I do not understand what the authors tried to study. The authors need to include the answers in the manuscript.
Major comments
#1. The authors answered that the hypothesis is described with the aim at the end of introduction. The hypothesis and the primary outcome are NOT clear (Page 2, Line 66). Describe the study hypothesis and the primary outcome more PRECISELY.
#2. Why did the authors exclude pain intensity less than 3/10 (Page 2, Line 77)? Include the answer (It is because generably this group of patients do not required changes in pain therapy) in the manuscript.
Minor comments
#1. What is “Karnofsky” (Table 1)?
Include the answer “Karnofsky is well know toll for measuring the level of physical activity in cancer patients” in the manuscript.
#2. The authors need to include at least body mass index (Table 1).
#3. Include explanations of Table 1.
Comments on the Quality of English LanguageNone.
Author Response
The authors do not respond my first comment. The study hypothesis and the primary outcome are not clear. I do not understand what the authors tried to study. The authors need to include the answers in the manuscript.
Major comments
#1. The authors answered that the hypothesis is described with the aim at the end of introduction. The hypothesis and the primary outcome are NOT clear (Page 2, Line 66). Describe the study hypothesis and the primary outcome more PRECISELY.
Done
#2. Why did the authors exclude pain intensity less than 3/10 (Page 2, Line 77)? Include the answer (It is because generably this group of patients do not required changes in pain therapy) in the manuscript.
Done
Minor comments
#1. What is “Karnofsky” (Table 1)?
Include the answer “Karnofsky is well know toll for measuring the level of physical activity in cancer patients” in the manuscript.
Done
#2. The authors need to include at least body mass index (Table 1).
As I answered before, we did not measure weight and height (I never seen this data in pain studies)
#3. Include explanations of Table 1.
Table 1. Epidemiological, clinical, and sociocultural characteristics of patients with low PCS (<30) and high PCS (≥30)
Round 3
Reviewer 4 Report
Comments and Suggestions for Authors
I have no further comments.
Comments on the Quality of English LanguageNone.